# A plant NLR receptor employs ABA central regulator PP2C-SnRK2 to activate antiviral immunity

Shen Huang[1], Chunli Wang[1], Zixuan Ding[1], Yaqian Zhao[1], Jing Dai[1], Jia Li [1], Haining Huang[1], Tongkai Wang[1], Min Zhu[1], Mingfeng Feng[1], Yinghua Ji[2], Zhongkai Zhang[3] & Xiaorong Tao [1]✉

Defence against pathogens relies on intracellular nucleotide-binding, leucine-rich repeat immune receptors (NLRs) in plants. Hormone signaling including abscisic acid (ABA) pathways are activated by NLRs and play pivotal roles in defence against different pathogens. However, little is known about how hormone signaling pathways are activated by plant immune receptors. Here, we report that a plant NLR Sw-5b mimics the behavior of the ABA receptor and directly employs the ABA central regulator PP2C-SnRK2 complex to activate an ABA-dependent defence against viral pathogens. PP2C4 interacts with and constitutively inhibits SnRK2.3/2.4. Behaving in a similar manner as the ABA receptor, pathogen effector ligand recognition triggers the conformational change of Sw-5b NLR that enables binding to PP2C4 via the NB domain. This receptor-PP2C4 binding interferes with the interaction between PP2C4 and SnRK2.3/2.4, thereby releasing SnRK2.3/2.4 from PP2C4 inhibition to activate an ABA-specific antiviral immunity. These findings provide important insights into the activation of hormone signaling pathways by plant immune receptors.

Defense against microbial pathogens relies on two tiers of innate immune system mediated by cell-surface pattern recognition receptors (PRRs) and intracellular nucleotide-binding domain-containing, leucine-rich repeat receptors (NLRs) in plants[1–3]. PRRs recognize microbe-associated molecular patterns and initiate pattern-triggered immunity (PTI)[4,5], whereas NLRs detect pathogen effectors that interfere with host PTI and activate a robust disease resistance known as effector-triggered immunity (ETI)[1–3]. The downstream immune outputs, including bursts of intracellular calcium, production of reactive oxygen species, and expression of defense genes, largely overlap in PTI and ETI[6,7]. Compared to PTI, which confers a transient, mild, and basal defense, the defense of ETI is often faster, stronger, and more prolonged[8]. Recent studies have shown that PTI can amplify and boost the defense signaling of ETI[9–12]. Despite this, the mechanism underlying how NLRs induce fast, strong, and prolonged defense in ETI remains largely unknown.

Hormone signaling pathways, including salicylic acid, jasmonic acid (JA), and ethylene signaling, are activated frequently by plant immune receptors[13,14]. These hormone signaling pathways play important roles in plant defense against different pathogens. Abscisic acid (ABA) is a major stress phytohormone that responds to both abiotic stress and biotic stress[15,16]. ABA was found to play a pivotal role in plant defenses against viruses, including Bamboo mosaic virus (BaMV), Tobacco necrosis virus (TNV), and Chinese wheat mosaic virus (CWMV)[17–19]. The ABA-insensitive mutant abi1-1 displayed greater susceptibility to BaMV[17]. Activation of the ABA signaling pathway was also found in Soybean NLR immune receptor Rsv3, which confers extreme resistance to Soybean mosaic virus (SMV); this activation of ABA signaling is important for callose deposition to plasmodesmata and inhibition of viral cell-to-cell movement[20]. Despite these hormone signaling pathways are known to play pivotal roles in plant defense,

[1]The Key Laboratory of Plant Immunity, Department of Plant Pathology, Nanjing Agricultural University, Nanjing 210095, P. R. China. [2]Institute of Plant Protection, Jiangsu Academy of Agricultural Sciences, Nanjing 210014, China. [3]Yunnan Academy of Tobacco Agricultural Sciences, Key Laboratory of Tobacco Biotechnological Breeding, National Tobacco Genetic Engineering Research Center, Kunming 650021, China. ✉e-mail: taoxiaorong@njau.edu.cn

little is known about how plant immune receptors, upon recognition of pathogens, initiate different hormone signaling pathways.

The major players of ABA signaling regulation include ABA receptors, protein phosphatases type-2C (PP2Cs), and sucrose-non-fermenting 1 (SNF1)-related kinases subfamily 2 (SnRK2s)[21,22]. The pyrabactin resistance (PYR)/pyrabactin resistance-like (PYL)/regulation components of ABA receptors (RCARs) are ABA receptor that belong to the START superfamily[23]. In the presence of ABA, the PYR/PYL/RCAR receptor complex undergoes conformational changes that enable binding to PP2C protein[24–26]. PP2C-type protein phosphatases are monomeric enzymes with large members in plants, and Group A PP2Cs function as negative regulators in ABA signaling[27,28]. SnRK2s phosphorylate and activate downstream factors[29]. However, group A PP2Cs interact with and constitutively inactivate SnRK2[29,30]. Thus, activation of SnRK2s is a key step in ABA signal transduction[31]. Upon the binding of ABA, PYR/PYL/RCAR receptors bind to PP2C[24–26], which competes with the interaction of SnRK2s with PP2C, thereby releasing SnRK2s from PP2C inhibition to activate downstream substrates and initiate ABA signaling[22].

*Tomato spotted wilt orthotospovirus* (TSWV), one of the most destructive plant viruses, has resulted in an annual loss of over $1 billion worldwide, posing a major threat to global crop productivity[32,33]. The most effective measure for controlling TSWV is to use of resistant cultivars. The *Sw-5b* is the most widely used, effective resistance gene in tomato-resistant cultivar breeding to control TSWV[34]. Sw-5b encodes an NLR immune receptor containing an N-terminal solanaceous-specific domain (SD), a coil-coil domain, an NB-ARC domain [nucleotide-binding (NB) adapters share sequence homology with Apaf-1, certain resistance proteins, and CED-4], and a leucine-rich repeat (LRR) domain[35,36]. Sw-5b induces a strong disease resistance against TSWV upon the recognition of viral effector NSm[37], a movement protein that facilitates viral intercellular movement through dilating the size limit of plasmodesmata in the plant cell wall[38]. Both SD and LRR domains are required for NSm recognition[39], whereas the NB domain of Sw-5b itself can trigger cell death, independent of the viral effector NSm[40].

In this study, we report that pathogen effector recognition by Sw-5b NLR induces an ABA-dependent antiviral immunity. Mechanically, we show that the Sw-5b NLR receptor mimics the behavior of the ABA receptors and employs the ABA central regulator PP2C4-SnRK2 complex to activate ABA signaling. Upon recognition of viral effector NSm, Sw-5b NLR undergoes conformational changes that allow it to bind to PP2C4 via the NB domain. The NBPP2C4 binding disrupts the interaction between PP2C4 and SnRK2.3/2.4, thereby releasing SnRK2 from PP2C inhibition for ABA-dependent defense against viral pathogens.

## Results

### The NB domain of Sw-5b interacts directly with group A PP2C4 and activates the ABA signaling pathway

Our previous results showed that the Sw-5b NB domain alone could induce elicitor-independent hypersensitive response (HR) cell death[40], suggesting that this domain plays a vital role in the activation of downstream immune signaling. To resolve the underlying mechanism, we used the NB domain as "bait" and performed yeast-two hybrid (Y2H) screening using the *Nicotiana benthamiana* cDNA library to identify the host proteins that interact with the Sw-5b NB domain. We identified 35 proteins as potential candidates, including SGT1, which has been shown to interact with other NLRs[41]. Among the selected NB-interacting proteins, one encoded *Nb*PP2C4 (Supplementary Table 1), a homolog of group A PP2C-type protein phosphatase (Supplementary Fig. 1) in *Arabidopsis* that functions in ABA signaling[22]. Y2H and split luciferase complementation (SLC) assays confirmed that Sw-5b NB interacts with *Nb*PP2C4 (Fig. 1a, b and Supplementary Fig. 2a). To test whether the interaction of NB with *Nb*PP2C4 can activate ABA signaling, we transiently expressed Sw-5b NB in *N. benthamiana* leaves. The

results showed that the representative ABA response genes were significantly upregulated upon expression of the Sw-5b NB domain (Supplementary Fig. 3a–c). ABA hormone levels were also significantly increased in plant leaves expressing Sw-5b NB (Supplementary Fig. 3d). The co-immunoprecipitation (Co-IP) assay results also showed that the full-length Sw-5b associated with the *Nb*PP2C4 protein only in the presence of the viral effector NSm (Fig. 1c). However, Sw-5b mutant with NB domain deletion did not interact with *Nb*PP2C4 (Supplementary Fig. 4a). Consistently, the auto-active Sw-5b$^{D857V}$ mutant[42] interacted with the *Nb*PP2C4 protein in the absence of NSm (Supplementary Fig. 4b). A quantitative reverse-transcription polymerase chain reaction (qRT-PCR) assay further demonstrated that the relative expression levels of ABA response genes were strongly upregulated in leaves co-expressing Sw-5b and NSm, but not in those expressing Sw-5b or NSm alone (Fig. 1d–f). ABA hormone levels were also significantly increased in plant leaves co-expressing Sw-5b and NSm (Fig. 1g). These data suggest that Sw-5b activates ABA signaling upon recognition of the viral effector NSm. We use tobacco rattle virus (TRV)-mediated gene silencing to knock down *NbABA2* and disrupt the ABA biosynthesis in *N. benthamiana* (Supplementary Fig. 5a–c), followed by co-expression of Sw-5b and NSm. The results showed that despite ABA biosynthesis is disrupted, co-expression of Sw-5b and NSm still significantly upregulates the ABA response genes (Supplementary Fig. 5d–f). The expression levels of the ABA response genes induced by Sw-5b in the *NbABA2*-silenced plants are lower than those in the non-silenced control plants (Supplementary Fig. 5d–f). Sw-5b localizes in both cytoplasm and nucleus[43]. To investigate which subcellular pools of Sw-5b can induce the ABA responses, we co-expressed NES-Sw-5b and NLS-Sw-5b with NSm in *N. benthamiana* leaves. The results showed that both the cytoplasmic and nuclear pools of the Sw-5b NLR can induce ABA response genes in the presence of NSm, whereas the nuclear pools of the Sw-5b induce slightly higher levels of ABA response genes than the cytoplasmic pools (Supplementary Fig. 6a–c).

Next, we examined the interaction between Sw-5b NB and *Sl*PP2C4 from tomato plants. The results of Y2H, Co-IP, and SLC assays showed that Sw-5b NB interacted with *Sl*PP2C4 in vitro and in planta (Fig. 1h, i and Supplementary Figs. 2b, 7). The full-length Sw-5b was also associated with *Sl*PP2C4 protein, but only in the presence of the viral NSm (Fig. 1j). The results of qRT-PCR assay further showed that the expression levels of the ABA response genes were significantly upregulated in tomato cv. IVF3545 plants (with *Sw-5b* gene) inoculated with TSWV (Fig. 1k–m). ABA levels were also significantly increased upon recognition of TSWV in tomato cv. IVF3545 plants (Fig. 1n). We also silenced *SlABA2* to inhibit ABA biosynthesis in tomato cv. IVF3545 plants (Supplementary Fig. 8a–c), followed by TSWV inoculation. TSWV inoculation still upregulates the ABA response genes in *SlABA2*-silenced plants compared to the mock-inoculated control plants (Supplementary Fig. 8d–f).

### Knockout/down *NbPP2C4/SlPP2C4* activates ABA-mediated defense against TSWV infection in *N. benthamiana* and tomato plants

As Sw-5b activates ABA signaling pathways, we first examined whether ABA signaling can provide a defense against TSWV infection. Wild-type (WT) *N. benthamiana* and tomato plants were treated with 100 μM ABA and mechanically inoculated with TSWV. Plants treated with 0.2% EtOH were used as a control. The results showed that disease symptoms and viral accumulation were significantly reduced in systemic leaves of WT *N. benthamiana* and tomato plants by ABA treatment (Fig. 2a–d and Supplementary Fig. 9a, b), suggesting that exogenous application of ABA induces defense against TSWV infection. To further investigate whether ABA-induced defense inhibits the replication or intercellular movement of TSWV, the leaves of *N. benthamiana* plants were treated with 100 μM ABA, followed by the inoculation of the

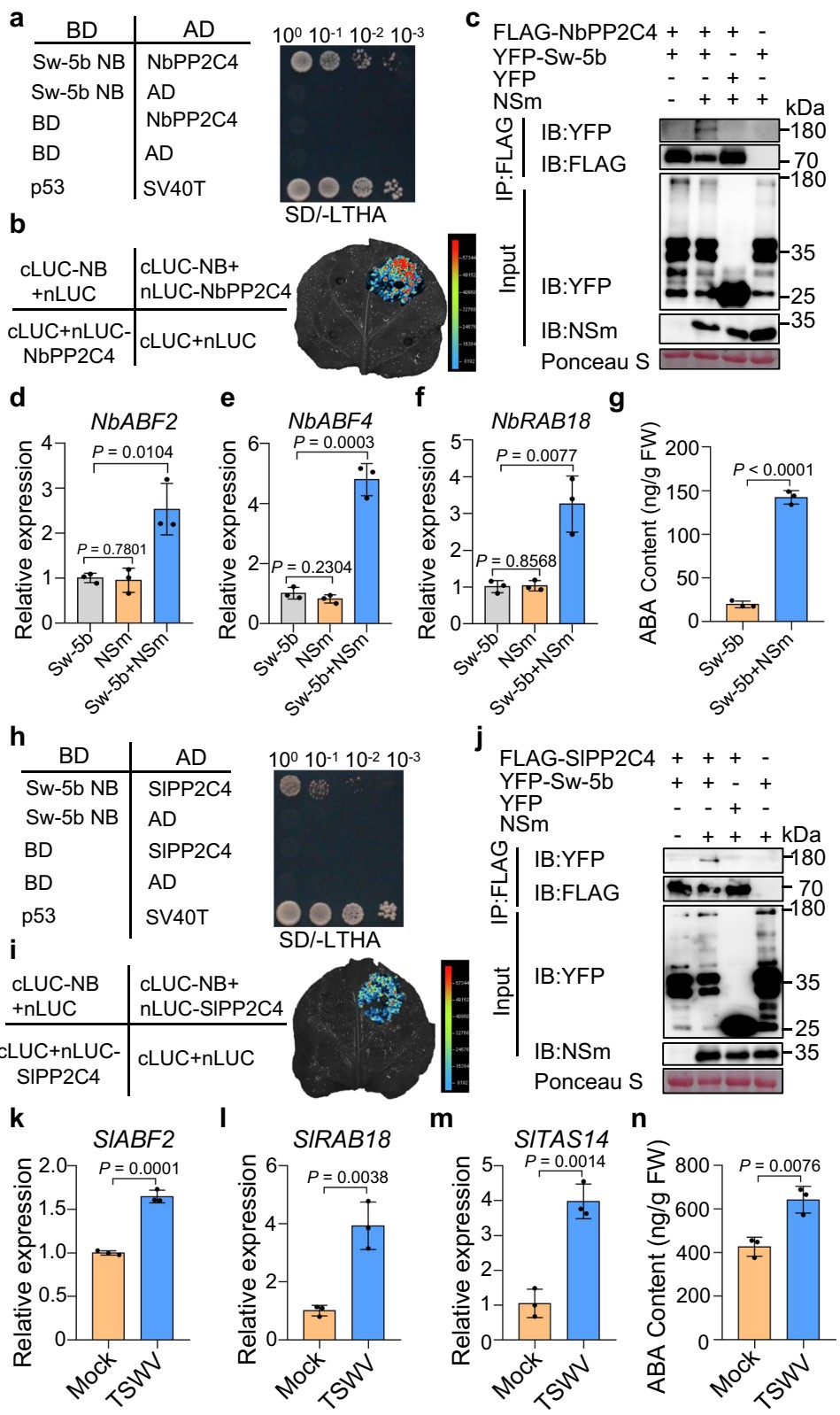

TSWV S segment-based minireplicon carrying eGFP reporter [SR$_{(+)}$eGFP] which can replicate only or the expression of the NSm movement protein::GFP fusion protein (NSm-GFP) which can move from one cell to other cells[43–45]. The results showed that ABA treatment not only inhibited the replication, but also disrupted the intercellular movement of TSWV (Supplementary Fig. 10a–d).

To further characterize PP2C4 in the ABA signaling pathway against viral infection, we generated two independent knockout mutant lines, *Nbpp2c4-5* and *Nbpp2c4-10*, using CRISPR-Cas9-based gene editing technology (Supplementary Fig. 11a, b). *Nbpp2c4-5* and *Nbpp2c4-10* mutants were more sensitive to ABA treatment in both seed germination and primary root growth inhibition assays compared

**Fig. 1 | Sw-5b NLR interacts directly with PP2C4 and activates ABA responses.**
**a** Y2H assay of the interaction between Sw-5b NB and NbPP2C4. AD activation
domain. BD DNA binding domain. BD-P53 and AD-SV40T were used as positive
controls. Protein interaction was assayed on SD/-LTHA dropout media. Yeast cell
growth in SD/-LT dropout media is shown in Supplementary Fig. 2a. **b** SLC assay of
the interaction between Sw-5b NB and NbPP2C4 at 48 h post-infiltration (hpi). **c** Co-
IP analysis of the interaction between Sw-5b and *Nb*PP2C4 with or without NSm.
Proteins in blots were detected with FLAG-, YFP-, or NSm-specific antibodies.
Ponceau S staining was performed to determine the amount of protein load. The
protein size marker is shown to the right. **d**–**f** Relative expression levels of *NbABF2*,
*NbABF4*, and *NbRAB18* in *N. benthamiana* leaves by (co)expressing Sw-5b, NSm or
both. **g** Amount of ABA in *N. benthamiana* leaves (co)expressing Sw-5b or Sw-5b

and NSm. The exact *P* value is provided in the Source Data. **h** Y2H assay for the
protein interaction between Sw-5b NB and *Sl*PP2C4 on SD/-LTHA dropout medium.
Yeast cell growth in SD/-LT dropout media is shown in Supplementary Fig. 2b. **i** SLC
analysis of the interaction between Sw-5b NB and *Sl*PP2C4 at 48 hpi. **j** Co-IP analysis
of the interaction between Sw-5b and *Sl*PP2C4. **k**–**m** Relative expression levels of
*SlABF2*, *SlRAB18*, and *SlTAS14* in tomato plant (+*Sw-5b*) leaves inoculated with TSWV
at 4 dpi. *SlActin* was used as the internal reference gene. Plants inoculated with
phosphate buffer (PB) were used as a Mock control. **n** Amount of ABA in tomato
plant (+*Sw-5b*) leaves inoculated with TSWV at 4 dpi. **d**–**g**, **k**–**n** Data were presented
as means ± s.d. (*n* = 3 biologically independent samples). Data were analysed by
two-sided Student's *t*-test. Source data are provided as a Source Data file. All
experiments were repeated at least three times with similar results.

to WT *N. benthamiana* plants (Fig. 2e and Supplementary Fig. 12a–c).
Consistently, the expression levels of ABA response genes *NbABF2*,
*NbABF4*, and *NbRAB18* (ABF, ABA-responsive element binding factor)
were greatly elevated in *Nbpp2c4* plants compared with those in WT
plants, in response to ABA treatment (Fig. 2f and Supplementary
Fig. 13a).

Next, we inoculated two *Nbpp2c4* mutant lines and WT *N. ben-
thamiana* plants with TSWV infectious clones (L_opt(+)+M_opt(-)+SR_(+)eGFP)
carrying enhanced green fluorescent protein (eGFP) reporter via
agroinfiltration[44]. Compared to the WT control plants, the *Nbpp2c4-5*
and *Nbpp2c4-10* mutant lines accumulated much less eGFP fluores-
cence expressed from the TSWV replicon (Supplementary Fig. 14a, b).
The *Nbpp2c4-5* and *Nbpp2c4-10* mutants were also mechanically
inoculated with TSWV-infected leaf extracts, resulting in a significant
reduction in disease symptoms and TSWV accumulation compared to
the WT control plants (Fig. 2g, h). To investigate whether this effect
was specific to TSWV, the leaves of *Nbpp2c4-5* and *Nbpp2c4-10* mutants
were also inoculated with an infectious clone of potato virus X carrying
a GFP reporter (PVX-GFP) and potato virus Y carrying a GFP repor-
ter (PVY-GFP), respectively, by agroinfiltration. The results showed
that both PVX-GFP and PVY-GFP accumulated significantly less in
*Nbpp2c4-5* and *Nbpp2c4-10* mutants than those in WT plants (Supple-
mentary Fig. 15a–d), suggesting that *Nbpp2c4* knockout also reduces
the infection of other viruses.

We also silenced *SlPP2C4* in cv. Moneymaker (without *Sw-5b*)
using TRV-based virus-induced gene silencing (VIGS). The expression
levels of ABA response genes *SlABF2*, *SlRAB18*, and *SlTAS14* were sig-
nificantly higher in *SlPP2C4*-silenced tomato plants than those in TRV-
GUS control plants in response to ABA treatment (Fig. 2i and Supple-
mentary Fig. 13b–d). *SlPP2C4*-silenced tomato plants were also
mechanically inoculated with TSWV-infected leaf extracts. As shown in
Fig. 2j, k, disease symptoms and viral accumulation were significantly
reduced in systemic leaves of *SlPP2C4*-silenced tomato plants com-
pared to control plants. Together, these results suggest that *NbPP2C4/
SlPP2C4* plays a negative role in the ABA response, and knockout/down
*NbPP2C4/SlPP2C4* activates the ABA-mediated defense against TSWV
infection in *N. benthamiana* and tomato plants.

**Overexpression of SnRK2.3/2.4 induces ABA-dependent immu-
nity to TSWV, and their function is inhibited by PP2C4**
SnRK2s are critical for ABA signal transduction[31]. Among ten *SnRK2s* in
*Arabidopsis*, *AtSnRK2.3*, *AtSnRK2.4*, and *AtSnRK2.6* are the major *SnRK2*
genes that participate in regulating ABA signaling[46]. *NbSnRK2.3/2.4* and
*SlSnRK2.3/2.4* from *N. benthamiana* and tomato are the closest
homologs to *AtSnRK2.3/2.4/2.6* (Supplementary Fig. 16a). Transient
overexpression of *NbSnRK2.3/2.4* or *SlSnRK2.3/2.4* upregulated the
expression of representative ABA response genes (Fig. 3a and Sup-
plementary Fig. 17a–d). Consistently, the expression levels of
*NbSnRK2.3* and *NbSnRK2.4* with TSWV infectious clones also inhibited
eGFP expression from the TSWV replicon in WT *N. benthamiana* leaves
(Fig. 3b). Similar results were observed for the expression levels of
*SlSnRK2.3* and *SlSnRK2.4* with the TSWV replicon (Supplementary

Fig. 18a). The inhibition of TSWV accumulation by *SnRK2.3/SnRK2.4*
was further confirmed by immunoblotting (Fig. 3c and Supplementary
Fig. 18b). However, silencing *NbSnRK2.3* and *NbSnRK2.4* using TRV-
based VIGS was lethal to *N. benthamiana* plants (Supplementary
Fig. 19), which suggests that NbSnRK2.3 and NbSnRK2.3 are required
for plant development and growth. Alternatively, we knocked down
*NbSnRK2.3/2.4* by transient expression of double-stranded hairpin RNA
(double-stranded RNA interference [dsRNAi]), which targeted the
mRNA of both *NbSnRK2.3* and *NbSnRK2.4* in WT *N. benthamiana* leaves.
Compared with dsGus control, leaves treated with dsNbSnRK2.3/2.4
had much higher eGFP expression from the TSWV replicon (Fig. 3d–f),
suggesting that NbSnRK2.3/2.4 play a positive role in the defense
against TSWV.

We also investigated the role of NbSnRK2.3/2.4 in Sw-5b-mediated
resistance against TSWV. When leaf halves from *NbSnRK2.3/2.4*-sup-
pressed or non-suppressed Sw-5b transgenic *N. benthamiana* plants
were rub-inoculated with TSWV-infected leaf extracts, TSWV inocula-
tion induced significantly more HR loci in *NbSnRK2.3/2.4*-suppressed
leaf halves than in non-suppressed leaf halves (Fig. 3g, h). Analysis of
TSWV-inoculated leaves through immunoblotting further showed that
the accumulation level of viral nucleocapsid protein in TSWV-
inoculated NbSnRK2.3/2.4-suppressed leaf halves was significantly
higher than that in TSWV-inoculated non-suppressed leaf halves
(Fig. 3i). To further investigate whether suppression of *NbSnRK2.3/2.4*
would affect the induction of cell death, we knocked down *NbSnRK2.3/
2.4* by transient expression of double-stranded hairpin RNA, followed
by co-expression of Sw-5b and NSm. We also treated *N. benthamiana*
leaves with 100 μM ABA as a control. The results showed that neither
*NbSnRK2.3/2.4* knockdown nor ABA treatment affected cell death
(Supplementary Fig. 20a–d).

The Y2H assay result showed that *Nb*PP2C4 interacted with
*Nb*SnRK2.3 and *Nb*SnRK2.4, respectively (Supplementary Fig. 16b).
Next, we tested whether *Nb*PP2C4 could inhibit the function of
*Nb*SnRK2.3/2.4 in activating immunity to TSWV by co-expressing
TSWV infectious clones carrying eGFP reporter with *NbSnRK2.3/2.4* and
empty vector (EV) control or *NbSnRK2.3/2.4* and *NbPP2C4* in *N. ben-
thamiana* leaves. Compared to the low GFP accumulation in the
*NbSnRK2.3/2.4* + EV control group, co-expression of *NbSnRK2.3/2.4*
with *NbPP2C4* significantly increased TSWV accumulation (Fig. 3j, k).
Similar results were observed by comparing the effects of *SlSnRK2.3/
2.4* and EV with *SlSnRK2.3/2.4* and *SlPP2C4* (Supplementary Fig. 21).
These results suggest that *NbPP2C4* inhibits the function of *NbSnRK2.3/
2.4* in activating the host defense against TSWV.

**Sw-5b NLR releases SnRK2.3/2.4 from PP2C4 inhibition via the
NB domain to activate antiviral immunity**
Given that the ABA-receptor complex PYR/PYL/RCAR in the presence
of ABA binds to PP2C and depresses PP2C repression of SnRK2s-
mediated activation of ABA signaling[47], we hypothesized that Sw-5b NB
may function in a similar manner to that of the ABA receptor and that
binding to PP2C4 may interfere with the interaction between PP2C4
and SnRK2s. Yeast three-hybrid (Y3H) assay results showed that

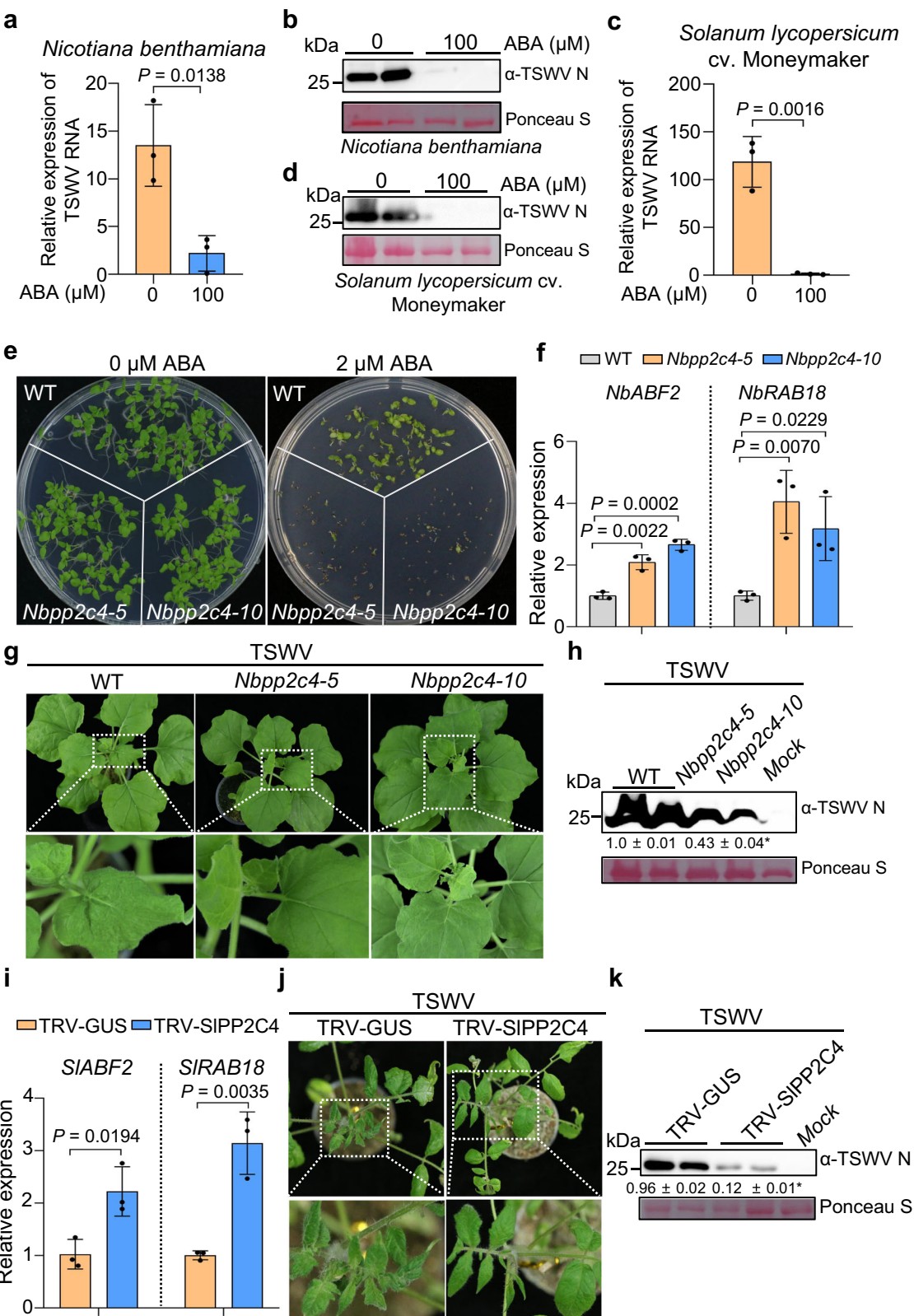

interaction between NbPP2C4 and NbSnRK2.3/NbSnRK2.4 was reduced significantly in the presence of Sw-5b NB compared to the yellow fluorescent protein (YFP) control (Fig. 4a). Glutathione S-transferase (GST) pull-down assays also showed that when the amount of YFP-Sw-5b-NB increased, the amount of FLAG-NbPP2C4 pulled down by GST-NbSnRK2.3/2.4 decreased (Fig. 4b, c). The same

assay showed that the addition of His-YFP had no effect on the interaction between FLAG-NbPP2C4 and GST-NbSnRK2.3/2.4 (Fig. 4b, c).

SlPP2C4 also interacted with SlSnRK2.3 and SlSnRK2.4, respectively, in yeast (Supplementary Fig. 16c). The Y3H assay results showed that the interaction between SlPP2C4 and SlSnRK2.3/SlSnRK2.4 was strongly compromised in the presence of Sw-5b NB, but not YFP

**Fig. 2 | Knockout/down *NbPP2C4/SlPP2C4* activates ABA-dependent defense against TSWV infection in *N. benthamiana* and tomato plants. a** Relative expression levels of TSWV RNAs in *N. benthamiana* plants with or without 100 μM ABA treatment. **b** Immunoblot analysis of TSWV accumulation in *N. benthamiana* plants with or without ABA treatment using viral N protein-specific antibodies. **c** Relative expression levels of TSWV RNAs in tomato plants (-*Sw-5b*) with or without ABA treatment. **d** Immunoblot analysis of TSWV accumulation in tomato plants (-*Sw-5b*) treated with or without ABA treatment. **e** Seed germination of *Nbpp2c4-5* and *Nbpp2c4-5* knockout mutants is inhibited in the presence of 2 μM ABA compared to wild-type (WT) *N. benthamiana* plants. Seeds were germinated on MS medium without ABA (left) or with 2 μM ABA (right) for 10 days. **f** Expression levels of *NbABF2* and *NbRAB18* in *Nbpp2c4-5* or *Nbpp2c4-10* mutant plants were significantly higher than that in WT *N. benthamiana* plants in response to ABA (100 μM). **g** Phenotypes of WT, *Nbpp2c4-5*, and *Nbpp2c4-10* mutant plants inoculated with TSWV photographed at 7 dpi. **h** Immunoblot analysis of TSWV accumulation in systemic leaves of WT, *Nbpp2c4-5*, and *Nbpp2c4-10* plants using viral N protein-specific antibodies. Plants inoculated with PB buffer were used as Mock controls. **i** Expression levels of *SlABF2* and *SlRAB18* in *SlPP2C4*-silenced tomato plants are significantly higher than those in TRV-*GUS* control plants in response to ABA (100 μM) treatment. **j** Phenotypes of TRV-*GUS* or TRV-*SlPP2C4*-treated tomato plants (-*Sw-5b*) infected with TSWV were photographed at 7 dpi. **k** Immunoblot analysis of TSWV accumulation in plant leaves shown in panel **j** using viral N protein-specific antibodies. Plants inoculated with PB were used as Mock controls. Ponceau S staining was used to determine the amount of protein loading. **a**, **c**, **f**, **i** Data were presented as means ± s.d. (*n* = 3 biologically independent samples). *P* values are indicated in the graphs or Source Data (two-sided Student's *t*-tests). **b**, **d**, **h**, **k** Experiments were repeated at least three times with similar results. Source data are provided as a Source Data file.

(Supplementary Fig. 22a). GST pull-down assays confirmed that the interaction between *Sl*PP2C4 and *Sl*SnRK2.3/*Sl*SnRK2.4 was reduced significantly with an increase in the amount of YFP-Sw-5b-NB; however, this interaction was not affected by the presence of YFP (Supplementary Fig. 22b, c).

Because Sw-5b NLR interacts with NbPP2C4 upon recognition of the viral effector NSm (Fig. 1), we investigated whether Sw-5b NLR can depress *NbPP2C4* inhibition of *NbSnRK2.3/2.4* during viral infection. We co-expressed TSWV infectious clones with *NbSnRK2.3/2.4* and *NbPP2C4*, or *NbSnRK2.3/2.4*, *NbPP2C4* and *Sw-5b* in *N. benthamiana* leaves. Compared to the high level of GFP accumulation attained by co-expression of NbSnRK2.3/2.4 and NbPP2C4, the addition of Sw-5b to the NbSnRK2.3/2.4-NbPP2C4 complex significantly inhibited TSWV accumulation (Fig. 4d, e). Next, we co-expressed TSWV SR$_{(+)eGFP}$ minireplicon with *NbSnRK2.3/2.4* and *NbPP2C4* in the presence of *EV*, *Sw-5b*, or *Sw-5b + NSm* in *N. benthamiana* leaves. The results showed that the co-addition of Sw-5b with only the NSm effector inhibited the accumulation of TSWV SR$_{(+)eGFP}$ minireplicon (Supplementary Fig. 23a–d). We also co-expressed TSWV infectious clones with *SlSnRK2.3/2.4* and *SlPP2C4*, or *SlSnRK2.3/2.4*, *SlPP2C4* and *Sw-5b* in *N. benthamiana* leaves. Similar effects were observed (Supplementary Fig. 24a, b). These results suggest that Sw-5b depresses PP2C repression of SnRK2.3/2.4 and activates a strong defense against TSWV.

### Sw-5b NLR induces stomatal closure and drought resistance in *N. benthamiana* and tomato plants upon recognition of the viral effector NSm

Because co-expression of Sw-5b and NSm induced ABA signaling, we examined whether Sw-5b NLR induces stomatal closure in the presence of NSm. When Sw-5b was co-expressed with the viral effector NSm in *N. benthamiana* leaves, stomatal closure was frequently detected (Fig. 5a, b). However, this effect was not observed in leaves expressing Sw-5b or NSm alone. When *NbSnRK2.3/2.4* was knocked down, co-expression of Sw-5b and NSm did not lead to activation of ABA response genes and induction of stomatal closure (Supplementary Fig. 25a–e), suggesting that Sw-5b induced stomatal closure depends on *NbSnRK2.3/2.4*. We also examined whether Sw-5b NLR can induce drought tolerance upon the detection of NSm. In response to drought, *Sw-5b* transgenic *N. benthamiana* plants with TSWV inoculation showed significantly enhanced drought tolerance compared with Mock-inoculated plants (Fig. 5c). In tomato plants carrying the *Sw-5b* resistance gene, stomatal closure was also frequently detected in TSWV-inoculated leaves compared with mock-inoculated plant leaves (Fig. 5d, e), and drought tolerance was significantly enhanced in TSWV-inoculated tomato plants carrying *Sw-5b* (Fig. 5f). These results suggest that Sw-5b NLR not only induces ABA-dependent immunity to viral infection but also promotes drought resistance in plants upon recognition of the viral effector protein.

## Discussion

In this study, we demonstrated that Sw-5b NLR behaves in a similar manner as ABA receptors and directly employs the PP2C4-SnRK2 complex, the central regulator of ABA signaling, to activate antiviral immunity. We show that pathogen effector recognition by Sw-5b NLR induces ABA-dependent antiviral signaling. The NB domain of Sw-5b NLR directly interacts with group A NbPP2C4/SlPP2C4 from *N. benthamiana* and tomato plants. Group A NbPP2C4/SlPP2C4 interacts with and inhibits NbSnRK2.3/2.4 or SlSnRK2.3/2.4. Knockout/down PP2C4 or overexpression of SnRK2.3/2.4 activates the ABA-specific defense pathway against TSWV infection in *N. benthamiana* or in tomato plants. Upon recognition of viral effector NSm, the Sw-5b NLR undergoes conformational changes that allow binding to PP2C4 via the NB domain. This NLR receptor-PP2C binding interferes with the interaction between PP2C4 and SnRK2.3/2.4, thereby releasing SnRK2.3/2.4 from PP2C4-mediated inhibition to activate the ABA-dependent defense against viral pathogens. Meanwhile, Sw-5b NLR-induced stomatal closure and drought resistance in *Nicotiana benthamiana* and tomato plants. These findings provide important insights into the activation of hormone signaling pathways by plant immune receptors.

The phytohormone ABA plays a pivotal role in plant defenses against viruses[48]. In this study, we found that Sw-5b NLR induces ABA accumulation and up-regulation of ABA response genes upon recognition of viral effector NSm. Exogenous application of ABA induces plant defense against TSWV infection. Consistently, knockout/down NbPP2C4/SlPP2C4 or overexpression of SnRK2.3/2.4 activates the ABA signaling−dependent defense against tospoviral infection (Fig. 2). These data suggest that the ABA signaling pathway activated by Sw-5b NLR plays a key role in plant defense against viral pathogens. Soybean NLR Rsv3-mediated extreme resistance against SMV was also found to induce ABA accumulation and downstream ABA signaling[20]. ABA was able to prime callose deposition and inhibit viral cell-to-cell movement[38]. The Sw-5b NLR-induced defense was also able to inhibit viral intercellular movement[43]. Activation of the ABA signaling pathway by both tomato Sw-5b and soybean Rsv3 NLRs suggests that ABA has an important antiviral role and that this signaling pathway can be employed by both NLRs to defend against viral infection.

Not only effector NSm recognition by Sw-5b NLR induces ABA accumulation and signaling, but also Sw-5b NB alone can induce ABA accumulation and signaling. Using Y2H screening, GST pull-down, co-IP, and SLC assays, we found that the Sw-5b NB domain directly interacts with group A NbPP2C4/SlPP2C4 from *N. benthamiana* and tomato plants (Fig. 1). Group A PP2Cs act in a negative feedback regulatory loop of the ABA signaling pathway[21,27,49]. Genetically, knockout/down *NbPP2C4/SlPP2C4* was more sensitive to ABA treatment. Meanwhile, the expression of ABA marker genes was much higher in *pp2c4* knockout mutants than in WT in response to ABA (Fig. 2), suggesting that *NbPP2C4/SlPP2C4* from *N. benthamiana* and tomato plants has a

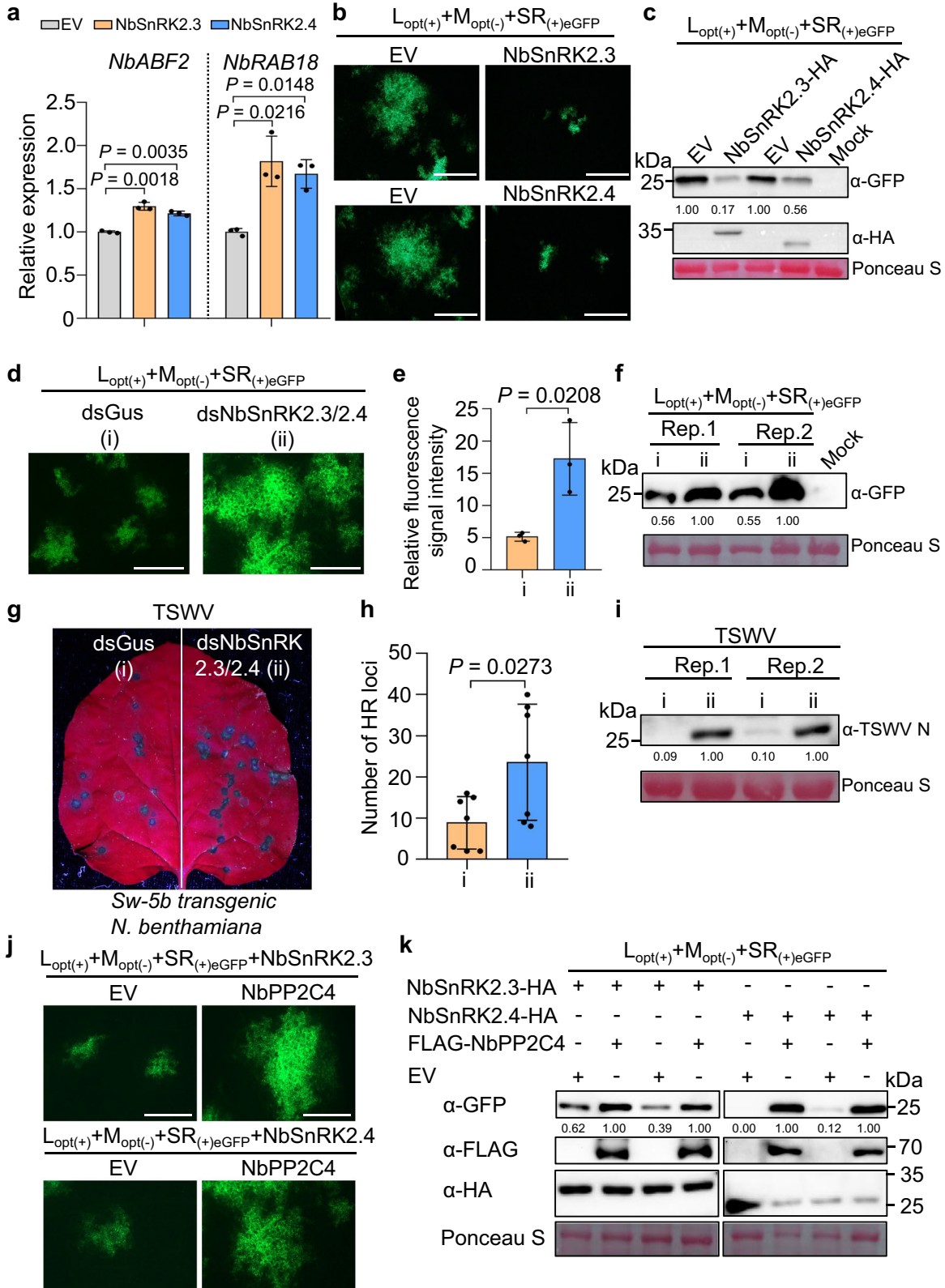

similar function to group A PP2Cs from Arabidopsis in ABA signaling. In Arabidopsis, PP2Cs physically interacted with SnRK2s and inactivated SnRK2[30]. Our Y2H and GST pull-down assays showed that NbPP2C4/SlPP2C4 interacts with and inhibits the function of NbSnRK2.3/4 or SlSnRK2.3/2.4 (Fig. 4a–c and Supplementary Fig. 22). SnRK2s are key regulators of ABA signaling transduction in Arabidopsis[46]. We found that overexpression of NbSnRK2.3/2.4 or

SlSnRK2.3/2.4, the orthologs of SnRK2s in Arabidopsis, significantly upregulated the expression of ABA-responsive genes (Fig. 3a and Supplementary Fig. 17), suggesting that SnRK2.3/2.4 from N. benthamiana and tomato plants play an important role in ABA signaling. Furthermore, the downregulation of SnRK2.3/2.4 reduces the Sw-5b-mediated disease resistance against virus infection, suggesting that SnRK2.3/2.4-mediated ABA signaling is required for Sw-5b to confer

**Fig. 3 | Overexpression of SnRK2.3/2.4 induces ABA-dependent immunity to TSWV, and their function is inhibited by PP2C4. a** Relative expression levels of *NbABF2* and *NbRAB18* in *N. benthamiana* leaves transiently overexpressing *Nb*SnRK2.3/2.4 at 24 h post-agroinfiltration (hpai). EV, p2300S empty vector. **b** Overexpression of *Nb*SnRK2.3/2.4 inhibits TSWV replicon accumulation in *N. benthamiana* plants. The eGFP fluorescence expressed from TSWV replicon (L$_{(+)opt}$, M$_{(-)opt}$, and SR$_{(+)eGFP}$) was examined using an inverted fluorescence microscope at 2.5 dpi. Bars = 800 μm. **c** Immunoblot analysis of TSWV accumulation in *N. benthamiana* leaves expressing NbSnRK2.3/2.4-HA using GFP and HA-specific antibodies, respectively. Mock, plant leaves treated with EV. **d** Silencing *Nb*SnRK2.3/2.4 enhanced TSWV accumulation in *N. benthamiana* leaves. The eGFP fluorescence expressed from the TSWV replicon was examined at 2.5 dpi. Bars = 800 μm. **e** Relative fluorescence signal intensity in leaves shown in panel **d** quantified using ImageJ software. **f** Immunoblot analysis of TSWV replicon accumulation in leaves shown in panel **d** with GFP-specific antibodies. **g** Silencing *NbSnRK2.3/2.4* reduced

*Sw-5b*-mediated resistance against TSWV in *N. benthamiana* plants. Sw-5b transgenic *N. benthamiana* plants treated with double-strand RNA targeting *NbSnKR2.3/2.4* (ds*NbSnKR2.3/2.4*) or with dsGus control were mechanically inoculated with TSWV-infected leaf sap. HR loci induced in inoculated leaves were monitored and photographed at 3 dpi. **h** Statistical analysis of HR loci treated with ds*NbSnKR2.3/2.4* or dsGus control in panel **g**. **i** Immunoblot analysis of TSWV accumulation in inoculated leaves shown in panel **g** using viral N protein-specific antibodies. **j** Overexpression of SnRK2.3/2.4 induces ABA-dependent immunity to TSWV, and their function is inhibited by PP2C4. The eGFP expressed from the TSWV replicon was examined at 2.5 dpi. Bars = 800 μm. **k** Immunoblot analysis of TSWV accumulation in panel **j** using GFP, FLAG, and HA-specific antibodies, respectively. Ponceau S staining was used to determine the amount of protein loading. **a**, **e** Data were presented as means ± s.d ($n = 3$); **h** $n = 7$. $P$ values are shown in the graphs (two-sided Student's *t*-test). Source data are provided as a Source Data file. All experiments were repeated at least three times with similar results.

full immunity. Importantly, Sw-5b NB interferes with the interaction between PP2C4 and SnRK2.3/2.4, thereby activating the SnRK2.3/2.4 to initiate the ABA signaling pathway in *N. benthamiana* and tomato plants to defend against TSWV.

We found that Sw-5b NLR interacts strongly with PP2C4 only when Sw-5b recognizes the viral effector NSm (Fig. 1), suggesting that the NB domain of Sw-5b NLR undergoes a major conformation change upon effector recognition and allowing NB binding to PP2C4. In our previous studies, we showed that the binding of the effector NSm onto the four polymorphic sites of the Sw-5b LRR domain interferes with the adjacent R927 residue and disrupts the intramolecular interaction between the NB and LRR domains, which in turn causes the conformational change of NB domain, thereby switching the Sw-5b immune receptor from an inactive state to an active state[37]. In the recently resolved 3D cryo-electron microscopy structure of the inactive and active states of the *Arabidopsis* NLR ZAR1, a key difference between the inactive and active states of ZAR1 is the conformational change in the NB domain[50]. The Sw-5b NB domain interacted directly with PP2C4 and the full-length of Sw-5b Sw-5b interacted with PP2C4 only in the presence of NSm. Our results, together with previous findings, suggest that the NB domain of Sw-5b undergoes a conformation change from an inactive state to an activated state upon effector recognition, and only with this activated conformation can Sw-5b NB associate with PP2C and activate the ABA signaling pathway.

Sw-5b plants exhibited significantly better drought tolerance with TSWV inoculation. Thus, as tomato plants evolve to the point of acquiring Sw-5b NLR, they may simultaneously acquire drought resistance capability. This hypothesis has important implications, in that Sw-5b NLR, a disease-resistance gene, may play both a virus-resistance role and a previously unknown role in drought resistance. Notably, plants carrying NLR experience a growth penalty, with ~5% yield loss[51] due to low levels of auto-activation of NLR. It is not known whether plants become more drought-tolerant when NLRs are introduced to plants. Whether Sw-5b or other NLRs are involved in both disease resistance and drought resistance, and whether these NLRs will be useful in breeding for both disease- and drought-resistant lines, are research avenues worthy of further exploration.

Based on the findings described above, we proposed a working model that Sw-5b, an NLR receptor of the plant innate immune system, mimics the behavior of the ABA receptors and directly modulates the ABA central regulator PP2C4-SnRK2 complex to activate ABA-dependent antiviral immunity (Fig. 6). At the center of the ABA signaling regulation, PP2C interacts with and constitutively inhibits SnRK2 function in the absence of ABA, whereas upon detection of ABA, the ABA receptor PYLs undergo a conformational change that enables binding to PP2C. The ABA-receptor binding releases SnRK2 from PP2C-mediated inhibition, thereby allowing SnRK2 to activate ABA signaling. The Sw-5b NLR immune receptor behaves in a similar manner to ABA receptors. In the absence of the viral effector NSm, Sw-5b NLR is

maintained in an inactive state. Upon recognition of the effector NSm, Sw-5b NLR switches from an inactive to an active conformation that allows binding to PP2C4 via the NB domain. The NLR-PP2C binding interferes with the interaction between PP2C4 and SnRK2.3/2.4 and releases SnRK2s from PP2C4 inhibition, thereby activating ABA-dependent antiviral immunity.

## Methods

### Plant material and growth conditions

The 6-to-8-week-old *Nicotiana benthamiana* plants were used for viral infection clone inoculation, protein analyses, and other experiments. The source of *Sw-5b* transgenic *N. benthamiana* lines has been described previously[42]. *Nbpp2c4* knockout *N. benthamiana* mutant lines were generated using a CRISPR/Cas9-based technology in this lab. T2 transgenic lines were used in this study. Seeds of tomato cv. 'IVF3545' carrying the *Sw-5b* resistance gene were provided by Prof. Junming Li at the Institute of Vegetables and Flowers, Chinese Academy of Agricultural Sciences, Beijing, China. Seeds of tomato cv. 'Moneymaker' without *Sw-5b* were provided by Dr. Hui Zhang at the Institute of Horticulture Science, Shanghai Academy of Agricultural Sciences, Shanghai, China. We used 4-week-old tomato plants for VIGS assays. All *N. benthamiana* plants and tomato seedlings were grown in an environmentally controlled greenhouse under a 16-h light (25 °C)/8-h dark (23 °C) photoperiod.

### Plasmid construction

The plasmids p2300S-YFP, p2300S-YFP-Sw-5b, p2300S-NSm, p2300S-Sw-5b, and p2300S-YFP-NB were described previously[40,42]. The coding sequences of PP2C4, SnRK2.3, and SnRK2.4 from *N. benthamiana* and tomato (*Solanum lycopersicum*) were amplified by RT-PCR, fused with HA or FLAG tags in their N-terminus or C-terminus, and cloned individually into pCambia2300S under the control of the 35S promoter[52].

To generate constructs expressing GST-SnRK2.3 and GST-SnRK2.4, their coding sequences were amplified by PCR and cloned into the pGEX-2TK vector. FLAG-PP2C4, YFP, and YFP-NB were cloned into pET-28a to generate pET-28a-FLAG-PP2C4, pET-28a-YFP, and pET-28a-YFP-NB, respectively.

AD-PP2C4, BD-SnRK2.3, BD-SnRK2.4, and BD-Sw-5b-NB were amplified by PCR and cloned into pGADT7 or pGBKT7. To generate the constructs BD-SnRK2.3-NB/YFP and BD-SnRK2.4-NB/YFP for the Y3H assays, Sw-5b-NB and YFP were amplified by PCR and cloned into a pBridge vector (Clontech, Mountain View, CA, USA) driven by a MET25 promoter. Subsequently, SnRK2.3 and SnRK2.4 were cloned into pBridge-NB/YFP, respectively.

The constructs for creating *Nbpp2c4* CRISPR-Cas9 knockout lines were generated by inserting a small guide RNA oligomer into the binary vector BGK01[53]. The synthesized sgRNA oligomers were annealed and then ligated to BGK01 with T4 DNA ligase to generate BGK01-NbPP2C4.

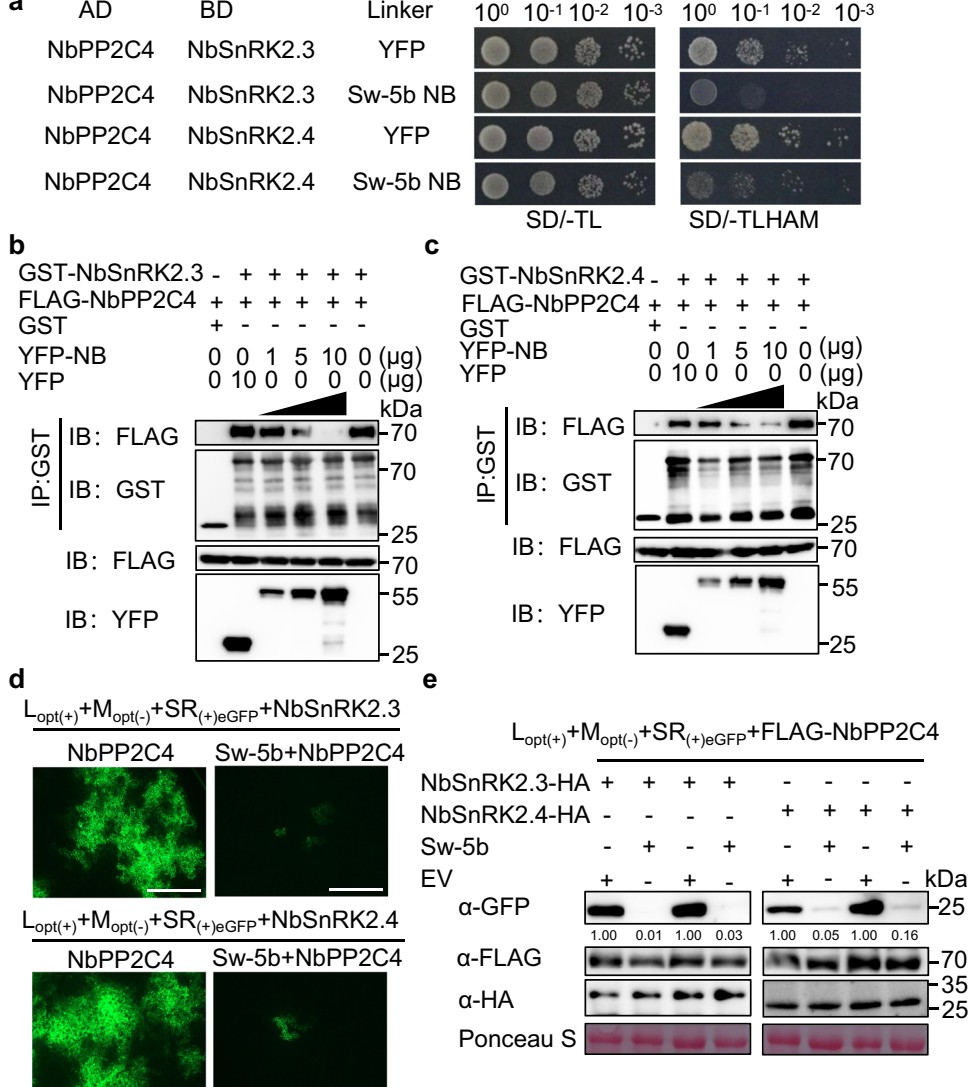

**Fig. 4 | Sw-5b NLR depresses PP2C4 repression of SnRK2.3/2.4 via the NB domain to activate ABA-dependent defense against TSWV. a** Yeast three-hybrid (Y3H) assays showing the effect of Sw-5b NB on the interaction between *Nb*PP2C4 and *Nb*SnRK2.3 or between *Nb*PP2C4 and *Nb*SnRK2.4. Yeast cells were co-transformed with pGAD-*Nb*PP2C4 and pBridge-*Nb*SnRK2.3/2.4 + YFP or pGAD-*Nb*PP2C and pBridge-*Nb*SnRK2.3/2.4 + Sw-5b NB domain. The interactions were assayed on SD/-TL (lacking Trp and Leu) and SD/-TLHAM (lacking Trp, Leu, His, Ade, and Met) dropout medium plates, respectively, for 3–5 days. **b, c** Glutathione *S*-transferase (GST) pull-down assays to assess Sw-5b NB interference with interaction between *Nb*PP2C4 and NbSnRK2.3/2.4 in vitro. Purified GST-NbSnRK2.3 (**b**) or GST-NbSnRK2.4 (**c**) protein was used to pull down the FLAG-NbPP2C4 with the

increasing amount of YFP-Sw-5b NB. YFP was used as a negative control. Blots were detected with FLAG, GST, and YFP-specific antibodies. The protein size marker is shown on the right. **d** TSWV L$_{(+)opt}$ + M$_{(-)opt}$ + SR$_{(+)eGFP}$ replicons with NbSnRK2.3 and NbPP2C4 (upper) or NbSnRK2.4 and NbPP2C4 (lower) were co-expressed with Sw-5b in *N. benthamiana* leaf halves. These combinations were also co-expressed with a p2300S EV in the other half of each leaf, which were used as controls. The eGFP fluorescence expressed from the TSWV replicon was examined and photographed under an inverted fluorescence microscope at 2.5 dpi. Bars = 800 μm. **e** Immunoblot analyses of TSWV replicon accumulation shown in panel **d** using GFP, FLAG, and HA-specific antibodies, respectively. All experiments were repeated at least three times with similar results.

To knock down SlPP2C4 in tomato plants, 300 bp of the coding sequence SlPP2C4 was cloned into pTRV2 to generate pTRV2-SlPP2C4. To knock down NbSnRK2.3 and NbSnRK2.4 simultaneously in *N. benthamiana* plants, partial sequences (250 bp) of NbSnRK2.3 and NbSnRK2.4 cDNAs were amplified by RT-PCR. The resulting fragments were fused together by overlap PCR and cloned into pTRV2 to produce pTRV2-NbSnRK2.3/2.4. To transiently silence NbSnRK2.3/2.4 genes by double-strand RNA in *N. benthamiana* leaves, the NbSnRK2.3/2.4 fused fragment was forward-inserted into the p2300s-intron vector cut with *Xba* I and *Pst*I restriction enzymes. The obtained construct was then reverse-inserted with an NbSnRK2.3&2.4 fusion fragment using *Kpn* I and *Bam*H I to generate p2300S-dsNbSnRK2.3/2.4.

To generate the cLUC-NB and nLUC-PP2C4 constructs, the coding sequences of Sw-5b-NB and PP2C4 were amplified by PCR, and cloned into the *35Spro*: cLUC and *35Spro*: nLUC vectors[54].

All primers used in this study are listed in Supplementary Table 2, and all constructs were verified by sequencing.

## Y2H screening and assays

Y2H screening was performed as reported previously[55]. Briefly, the Sw-5b-NB were cloned into the pGBKT7 vector (Clonetech) and transformed into the Y2H-gold yeast strain. Yeast cells containing pGBKT7-Sw-5b-NB were mated with the *N. benthamiana* cDNA prey library, which was constructed on the previously described pGADT7 prey vector[55] at 30 °C and 50 rpm for 20 h. The fusion cells were incubated

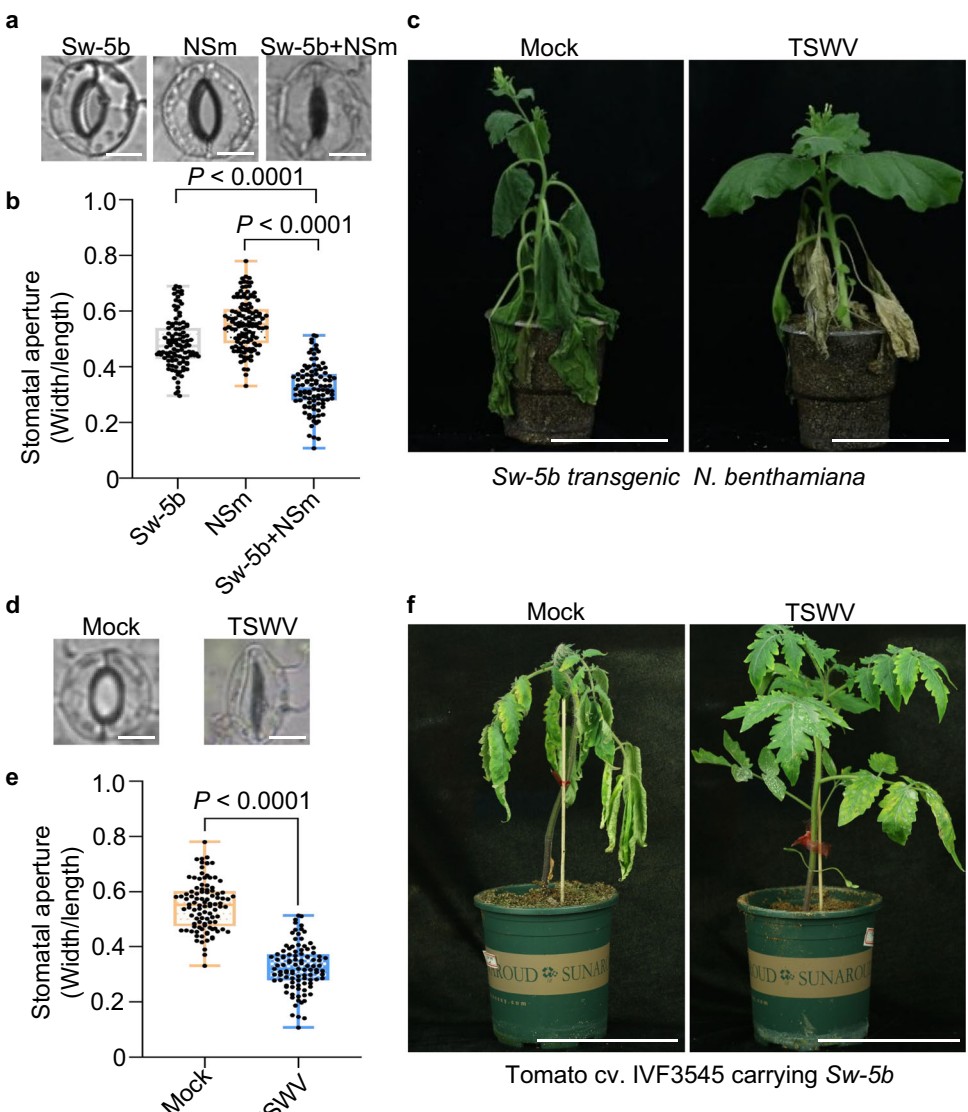

**Fig. 5 | Sw-5b NLR induces plant stomatal closure and drought resistance in *N. benthamiana* and tomato plants upon recognition of viral effector NSm. a** Co-expression of Sw-5b and NSm induces stomatal closure in *N. benthamiana* leaves. Sw-5b, NSm or Sw-5b and NSm were (co)expressed in *N. benthamiana* leaves via agroinfiltration. The stomatal aperture was measured at 24 hpi. Bars = 10 μm. **b** Statistical analysis results of stomatal aperture in leaves treated with Sw-5b, NSm, or Sw-5b and NSm via agroinfiltration. **c** *Sw-5b*-transgenic *N. benthamiana* plants inoculated with TSWV-infected leaf sap were assayed for drought resistance through water shortage treatment for 10 days. Plants rub-inoculated with sap from healthy plant leaf tissues were used as mock controls. Bars = 8 cm. Experiments were repeated three times with similar results. **d** Stomatal closure analysis of tomato

leaves (+Sw-5b) inoculated with TSWV or mock control at 4 dpi. Bars = 10 μm. **e** Statistical analysis results of stomatal aperture in tomato leaves treated with TSWV or mock control, as shown in panel **d**. **f** Tomato plants carrying the *Sw-5b* resistance gene inoculated with TSWV-infected leaf sap were assayed for drought resistance through water shortage treatment for 10 days. Plants rub-inoculated with sap from healthy plant leaf tissues were used as mock controls. Bars = 12 cm. **b, e** Data were shown as the box plots with the interquartile range as the upper and lower confines, minima and maxima as whiskers, and the median as a solid line (*n* = 100, the number of stomata); the exact *P* value (two-sided Student's *t*-test) is provided in Source Data. Source data are provided as a Source Data file. All experiments were repeated at least three times with similar results.

at 28 °C on an SD/-TLHA medium plate for 5 days. Positive prey clones grown on SD/-TLHA medium were propagated in Leu-deficient medium to extract the plasmid and co-transformed into Y2H-gold yeast cells together with pGBKT7-Sw-5b-NB to confirm their interactions. The insert of the positive clones that interacted with Sw-5b NB was sequenced and analyzed by BLASTn.

For the Y2H assays, gold yeast strain cells were transformed with various combinations of BD- and AD-derived constructs and were grown at 28 °C for 3 days on the SD/-Trp-Leu medium. The yeast cells were diluted in ddH₂O at ratios of 1:1, 1:10, 1:100, and 1:1000 (v/v) and grown on SD/-Trp-Leu and SD/-Trp-Leu-His-Ade media for 3 days at 28 °C. Yeast transformed with BD-P53 and AD-SV40T was used as a positive control.

### SLC assay

Agrobacterium-mediated transient expression in *N. benthamiana* leaves was performed following the method of ref. 39. All constructs used in this study were transformed into *Agrobacterium tumefaciens* strain GV3101, grown to an optical density at 600 nm (OD₆₀₀) of 0.8–1.0, centrifuged at 6000 rpm for 10 min, and then resuspended with an infiltrating buffer (10 mM MES [pH 5.6], 10 mM MgCl₂, and 150 mM acetosyringone) to OD₆₀₀ = 0.2. The cell suspension mixture was infiltrated into the abaxial (lower) side of *N. benthamiana* leaves using a 2-mL needleless syringe.

The SLC assay was performed according to the method of Chen et al. with slight modifications[54]. *Agrobacterium tumefaciens* GV3101 strains carrying different combinations of constructs expressing

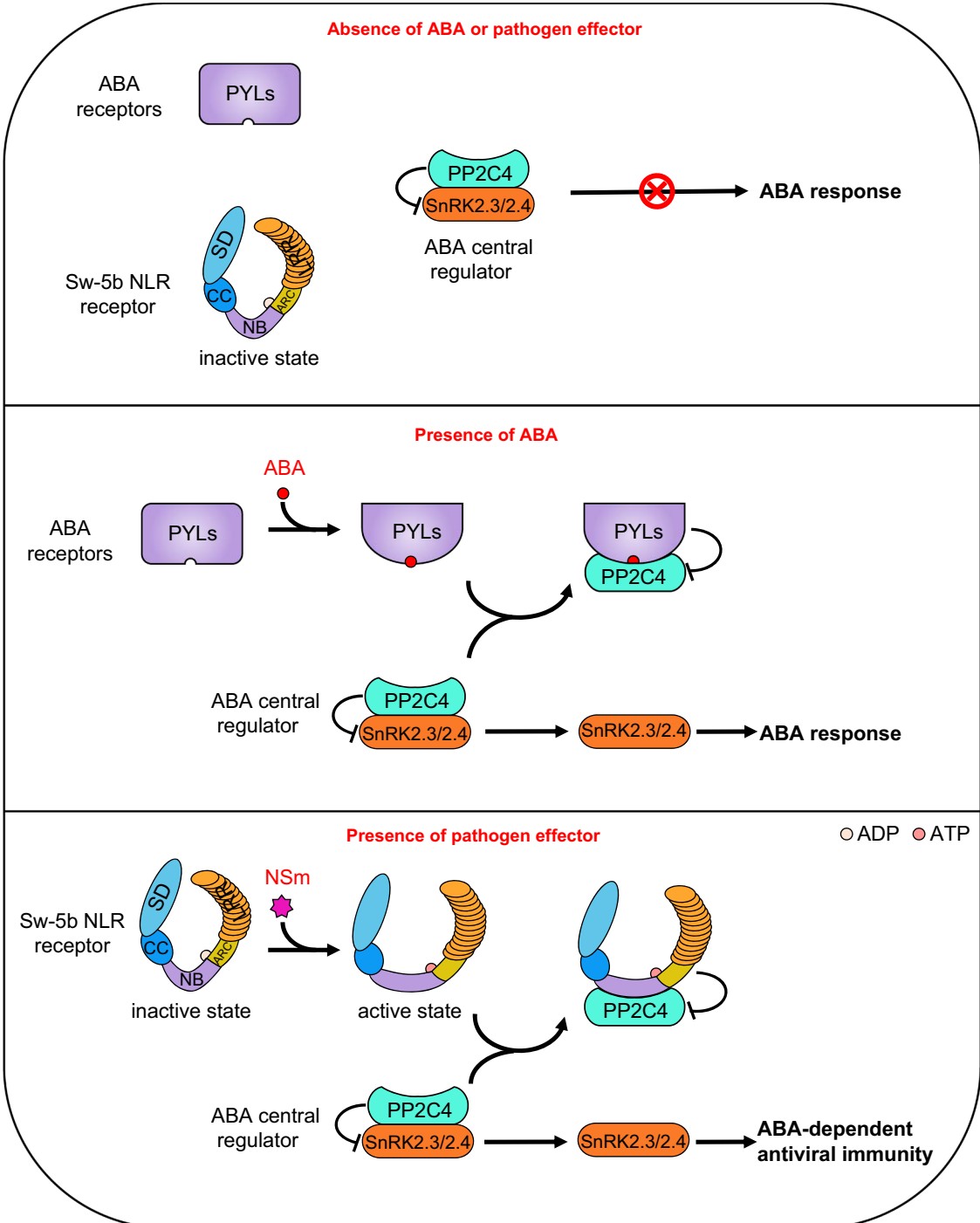

**Fig. 6 | A proposed model that the Sw-5b NLR immune receptor mimics the behavior of the ABA receptors and directly modulates the ABA central regulator PP2C-SnRK2 to activate antiviral immunity.** At the center of the ABA signaling regulation, PP2C4 interacts with and constitutively inhibits SnRK2.3/2.4. Upon detection of ABA, the ABA receptors PYLs undergo a conformational change that allows them to bind to PP2C4. This binding releases SnRK2 from PP2C inhibition, thereby allowing SnRK2 to activate the downstream ABA signaling pathway (upper). The Sw-5b NLR immune receptor behaves in a similar manner to ABA receptors to activate ABA signaling. Sw-5b NLR is maintained in an inactive state in the absence of the viral effector NSm. Upon detection of the viral effector NSm, Sw-5b NLR switches from an inactive state to an active state. The conformational changes of Sw-5b allow it to bind to PP2C4 through its NB domain, and this binding interferes with the interaction between PP2C4 and SnRK2.2/2.4, thereby releasing SnRK2 from PP2C-mediated inhibition to activate ABA-dependent antiviral immunity (lower).

cLUC-Sw-5b NB, nLUC-PP2C4, cLUC, or nLUC were infiltrated into *N. benthamiana* leaves. After 36 h of infiltration, the leaves were treated with 150 µg/mL of luciferin potassium salt (Yeasen, Shanghai, China) in the dark for 15 min. Luciferase activity was captured by a low-light cold charge-coupled imaging apparatus (Vilber, Ile-de-France, France).

### Antibody preparation
For the detection of NSm and N proteins, a 6×His tag was fused with the proteins, expressed in *E. coli* Rosetta (DE3) cells, and purified using Ni-NTA agarose (Qiagen). The purified NSm or N protein was injected into rabbits to produce polyclonal antibodies.

## Co-IP assays

For Co-IP assays, 1 g of agro-infiltrated *N. benthamiana* leaves was harvested and ground in 2 mL of chilled extraction buffer (10% [v/v] glycerol, 25 mM Tris-HCl [pH 7.5], 1 mM EDTA, 150 mM NaCl, 10 mM dithiothreitol, 2% [w/v] polyvinylpolypyrrolidone, 0.5% [v/v] Triton X-100, and protease inhibitor cocktail) using a mortar and pestle. The mixture was centrifuged at $18,000 \times g$ for 30 min at 4 °C. The supernatant was incubated with 30 μL of anti-FLAG (Sigma-Aldrich, St. Louis, MO, USA) agarose beads for 2 h at 4 °C. After incubation, the beads were washed four times with IP (50 mM Tris−Cl [pH 8.5], 100 mM NaCl, and 1 mM EDTA) buffer, resuspended in sodium dodecyl−sulfate-polyacrylamide gel electrophoresis (SDS−PAGE) loading buffer (150 mM Tris-HCl, pH = 6.8, 30% glycerol, 6% SDS, 0.3% bromophenol blue, and 300 mM DTT), and boiled at 95 °C for 10 min. The proteins were separated by SDS−PAGE and transferred to polyvinylidene fluoride membranes. The blots were detected using anti-YFP (Sigma-Aldrich, Cat. # SAB4301138, 1:10,000), anti-FLAG (Sigma-Aldrich, Cat. # A8592, 1:10,000), and anti-NSm (made in this study, 1:5,000) followed with Goat Anti-Rabbit IgG-HRP secondary antibody (Sigma-Aldrich, Cat # A0545, 1:10,000). The blots were stained with Ponceau S to estimate protein loading. The uncropped blots are provided in the Source Data file.

## qRT-PCR

Total RNA was extracted from *N. benthamiana* and tomato leaves using a FastPure Universal Plant Total RNA Isolation Kit (Vazyme Biotech, Nanjing, China) according to the manufacturer's instructions. The extracted RNA was reverse-transcribed into cDNA by HiScript III RT SuperMix (Vazyme Biotech). We performed qRT-PCR using ChamQ Universal SYBR qPCR Master Mix (Vazyme Biotech) on an ABI Prism 7500 Fast Real-Time PCR system (Life Technologies, Beverly, MA, USA). *NbActin* and *SlActin* were used as internal controls. Primers were designed using Primer Premier 5 software and are listed in Supplementary Table 2.

## ABA hormone measurement

Fresh plant material (100 mg) was ground into powder using liquid nitrogen in a mill. The powdered sample was homogenized in 1 mL of extraction buffer (isopropyl alcohol:water:hydrochloric acid, 2:1:0.002) and incubated at 4 °C for 1 h. Dichloromethane (2 mL) was added to the mixture, which was then incubated for 30 min. After centrifugation at $18,000 \times g$ for 15 min at 4 °C, the organic layer of the solution was absorbed and dried through evaporation under a nitrogen gas stream. The dried powder was reconstituted in methanol, and the solution was filtered through a 0.22-μm filter. The ABA content of samples was analyzed using a Triple Quadrupole Xevo TQ-S System (Waters Corp., Milford, MA, USA) equipped with an ESI ion source and a Waters ACQUITYUPLCBEH C18 column (2.1 mm YUPLCB, 1.71 mm).

## Generation of *Nbpp2c4* knockout *N. benthamiana* lines using a CRISPR/Cas9-based technology

*Nbpp2c4* knockout mutants were generated using CRISPR/Cas9-based technology following the method of ref. 55. Briefly, leaf disks from 8-week-old *N. benthamiana* plants sterilized with 70% ethanol and 10% sodium hypochlorite solution were incubated in a culture of *A. tumefaciens* carrying BGK01-NbPP2C4 for 15 min at 28 °C. Subsequently, the leaf disks were incubated on a co-cultivation medium for 2 days at 25 °C in the dark. After co-cultivation, the leaf disks were transferred to a shoot regeneration medium and incubated in a growth chamber for 15−20 days until the shoots reached a length of 1−2 cm. The shoots were then separated from the callus and transferred to a rooting medium for 20−30 days. When the roots reached a length of 3−5 cm, the plants were transferred into soil and grown in a greenhouse. Genomic DNA was extracted from regenerated plants, and the targeted sites of NbPP2C4 were PCR-amplified and sequenced to confirm

that the NbPP2C4 genes were indeed knocked out. Homozygous T2 lines were used in subsequent experiments.

## VIGS

VIGS was performed following the method of ref. 55. A mixture of Agrobacterium cells harboring both pTRV1 and pTRV2-NbSnRK2.3/2.4, pTRV2-NbABA2, pTRV2-SlPP2C4, pTRV2-SlABA2, or pTRV2-GUS was co-infiltrated into leaves of 4-week-old *N. benthamiana* or tomato plants. At 3 weeks post-agroinfiltration, the upper leaves of the infiltrated plants were analyzed for gene silencing efficiency. The leaves of plants with high silencing efficiency were used for TSWV inoculation and other experiments.

## Seed germination and root elongation assays

The effects of ABA on seed germination and root growth of *NbPP2C4* knockout mutant lines were evaluated following the method of Rubio et al. with slight modifications[56]. For seed germination assays, approximately 80 seeds per experiment were grown on Murashige and Skoog (MS) medium supplemented with or without 2 μM ABA (Sigma-Aldrich) in a growth chamber under a 16-h light (25 °C; white light, 4000 lumens)/8-h dark (23 °C) photoperiod. After 10 days of incubation, the seedling greening ratio was calculated, and the phenotypes were photographed.

For the root growth inhibition assay, surface-sterilized seeds were sown on MS agar medium supplemented with or without 0.3 μM ABA. Root elongation was measured and photographed after the plates were grown in a growth chamber for 15 days.

## TSWV mechanical inoculation and *Agrobacterium*-mediated inoculation of infectious clones

For TSWV mechanical inoculation, TSWV-infected *N. benthamiana* leaf tissues were harvested and ground in 1× phosphate buffer (137 mM NaCl, 2.7 mM KCl, 10 mM $Na_2HPO_4$, and 2 mM $KH_2PO_4$, pH 7.4). Leaf sap was rub-inoculated onto the leaves of 4−6-week-old *N. benthamiana* or tomato plants. Plants were either VIGS pretreated, transiently overexpressed with proteins, or treated with ABA. The inoculated plants were grown in a growth chamber under a 16-h light (25 °C)/8-h dark (23 °C) photoperiod.

*Agrobacterium*-mediated inoculation of infectious clones was conducted as previously described in ref. 42. A mixture of *A. tumefaciens* harboring the TSWV full-length infectious clones of $L_{(+)opt}$, $M_{(-)opt}$, and $SR_{(+)eGFP}$ was co-infiltrated into the leaves of 4−6-week-old *N. benthamiana* plants in various assays, as described above.

## Y3H assays

For Y3H assays, constructs of AD-PP2C4 and either BD-SnRK2.3-YFP/NB or BD-SnRK2.4-YFP/NB were co-transformed into the Y2H-gold yeast strain and incubated at 28 °C on SD/-Trp-Leu medium for 3 days. Individual colonies were picked and cultured in a liquid medium lacking Trp overnight. The yeast culture was centrifuged at 3000 rpm. The pellet was resuspended in ddH$_2$O and adjusted to OD$_{600}$ = 1.0. The yeast cells were then diluted at ratios of 1:1 (undiluted), 1:10, 1:100, and 1:1000 (v/v) with ddH$_2$O and plated on SD/-Trp-Leu and SD/-Trp-Leu-His-Ade-Met solid media at 28 °C for 3 days.

## GST pull-down assays

Proteins were expressed in the *E. coli* 'Rosetta' (DE3) strain and purified as previously described[39]. For the GST pull-down assays, the GST-SnRK2.3/2.4 proteins were mixed with FLAG-PP2C4 and incubated with 30 μL glutathione-agarose beads for 1.5 h at 4 °C. The beads were washed with pre-cold IP buffer (50 mM Tris−Cl [pH 8.5], 100 mM NaCl, 1 mM EDTA) and then incubated with 1, 5, or 10 μg YFP-NB, or 10 μg YFP protein at 4 °C for 1 h. After four washes with IP buffer, protein complexes bound onto the beads were boiled and separated by SDS−PAGE. The proteins were then transferred onto blots and detected with anti-

GST (Sigma-Aldrich, Cat. # SAB1305539, 1:5,000), anti-FLAG (Sigma-Aldrich, Cat. # A8592, 1:10,000), and anti-YFP (Sigma-Aldrich, Cat. # SAB4301138, 1:10,000) antibodies, followed with Goat Anti-Rabbit IgG-HRP secondary antibody (Sigma-Aldrich, Cat # A0545, 1:10,000).

**Stomatal aperture and drought resistance assays**
Stomatal apertures in epidermal strips peeled from *N. benthamiana* leaves transiently expressing NSm, Sw-5b, and Sw-5b+NSm, or tomato leaves inoculated with TSWV, were photographed and measured using ImageJ software (National Institutes of Health, Bethesda, MD, USA). Drought resistance assays were performed using 6-week-old *Sw-5b* transgenic *N. benthamiana* plants or 8-week-old tomato plants harboring the *Sw-5b* resistance gene. The *N. benthamiana* or tomato plants were rub-inoculated with sap from TSWV-infected *N. benthamiana* leaf tissues, whereas plants rub-inoculated with sap from healthy *N. benthamiana* leaf tissues were used as controls. The plants were then deprived of water for 10 days under the same growing conditions and subsequently photographed.

**Quantification and statistical analyses**
All statistical analyses were performed by two-sided Student's *t*-test using GraphPad Prism 9 (GraphPad Software, Boston, MA, USA). Data for quantification analyses are presented as means ± s.d. or as box plots with the interquartile range as the upper and lower confines, minima and maxima as whiskers, and the median as a solid line. Quantification analyses of protein abundance were conducted using ImageJ. Significance was determined at $P < 0.05$.

**Reporting summary**
Further information on research design is available in the Nature Portfolio Reporting Summary linked to this article.

## Data availability
All data were available within this article and its supplementary files. All constructs and transgenic plants are available upon request. The PP2C4 and SnRK2 gene sequences are available from the Sol Genomics Network (https://www.sgn.cornell.edu/) using the accession numbers provided in this article. Source data are provided with this paper.

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

## Acknowledgements
This work is supported by the National Natural Science Foundation of China (Grant Nos. 31925032, 32220103008, and 31972241), the National Key Research and Development Program of China (Grant No. 2022YFF1001500), the Funds from the Independent Innovation of Agricultural Science and Technology of Jiangsu Province [Grant No. CX (22) 2039], the Jiangsu Key Technology R & D Program and International Science and Technology Cooperation Project (Grant No. BZ2023030), the "111" project, the Key Science and Technology Program of Hainan Province (Grant No. ZDKJ2021007), the Guidance Foundation, the Sanya Institute of Nanjing Agricultural University (Grant No. NAUSY-MS19), and Yunnan Seed Laboratory (Grant No. 202205AR070001) to X.T., and the Postgraduate Research & Practice Innovation Program of Jiangsu Province (Grant No. KYCX21_0620) to S.H.

## Author contributions
S.H. and X.T. conceived and designed the experiments; J.L., H.H., M.Z., Y.J., Z.Z., and M.F. provided input. S.H., C.W., Z.D., Y.Z., J.D., and T.W. performed the experiments. S.H. and X.T. wrote the paper.

## Competing interests
The authors declare no competing interests.
