## [Peer Review File · Nature Communications]

REVIEWER COMMENTS

Reviewer #1 (Remarks to the Author):

In this manuscript the authors report Sw-5b NLR-mediated regulation of ABA to activate antiviral immunity against TSWV. Manuscript is well written and results are interesting.

Specific comments:

Authors found that NB domain of Sw-5b NLR interacts with NB domain through Y2H and then confirmed the interaction using split luciferase assay. Authors have also confirmed the interaction between full-length Sw-5b and PP2C only in the presence of NSm effector. Does full length Sw-5b interaction is affected if NB domain is deleted? Also, does auto-active Sw-5b interacts with PP2C in the absence of NSm?

Authors show that Nbpp2c knockout increases ABA responsive genes and reduces TSWV infection. Did author test if this effect is specific to TSWV or does it also affect other virus infection?

The data shown clearly demonstrate that the RNAi of SnRK2.3/2.4 enhances TSWV infection and induces more HR spots. Authors show that Sw-5b interferes with the interaction between PP2C and SnRK in the presence of TSWV. Does this interference occur only in the presence of NSm effector?

It is interesting that Sw-5b NLR induces stomatal closure and drought resistance. Does Sw-5b coexpression with NSm fail to induce stomatal closure in Nbpp2c knockout plants? Is drought resistance in Sw-5b line dependent on NbPP2C?

Reviewer #2 (Remarks to the Author):

The manuscript describes the link between the plant immune NLR receptor Sw-5b and the activation of an ABA-dependent antiviral response against TSWV infection. The authors demonstrate that ABA

application has a disease-suppressing effect and that the regulatory PP2C-SnRK2 complex involved in ABA signalling contributes to antiviral immunity. They further show that effector recognition by Sw-5b triggers the binding of Sw-5b to PP2C4 which releases SnRK2 from PP2C4 inhibition and leads to an activation of the ABA-dependent defences. Thus, the authors propose a model in that the Sw-5b mimics the action of ABA receptors that upon ABA binding disrupt the PP2C4-SNRK2 interaction to induce downstream ABA responses.

Together, this manuscript provides interesting mechanistic insight into immune-receptor activation of a hormone signalling pathways and adds to the previously described role of ABA in virus infection. There are, however, several issues that need to be addressed to further substantiate the main conclusions and consider the paper for publication.

#1) Fig. 1d-g/Suppl. Fig.3. Transient expression of the Sw-5b NB domain or full-length Sw-5b together with the TSWV effector NSm triggers ABA signalling and increases ABA levels. To distinguish that Sw-5b induces ABA signalling independent of ABA perception, the expression analysis of ABA response genes as well as TSWV accumulation needs to be done in an ABA deficient background, e.g. by disrupting ABA biosynthesis. The same applies to the experiment done in the tomato system (Fig 1k-n).

#2) Figure 3. It is quite puzzling that silencing of NbSnRK increases the number of HR lesions, which is associated with increased levels of TSWV. How has the virus level been determined, in total leaves or in the lesions? If a proper HR is induced, the virus should not spread beyond the lesions. Hence, it remains completely unclear if the Sw-5b activated ABA-dependent immunity branch is connected to or separated from the cell death-mediated resistance response. Does ABA effect the HR? Can the Sw-5b mediated ABA immunity be uncoupled from the HR? Is it the main component of the Sw-5b mediated resistance, or does it act additively to the cell death response? This part is particularly relevant to clarify experimentally as the previously observed extreme resistance to soybean mosaic virus (SMV) has been coupled to ABA mediated responses but occurs in the absence of HR.

#3) Based on the previous publication from the group (Chen et al., 2021, New Phytol.), it remains elusive if the ABA-mediated response is related to the cytoplasmic or nuclear pools of the NLR, in particular as they separate the effects of Sw-5b mediated responses on replication (cytoplasmic) and cell-to-cell/long-distance movement (nuclear). In case of the Rsv-mediated extreme resistance to SMV, ABA seems to effect in particular callose deposition at plasmodesmata to prevent intercellular virus spread. What is the mechanism of ABA-immunity in case of TSWV?

Response to reviewers:

1. Reviewer #1 (Remarks to the Author):

In this manuscript the authors report Sw-5b NLR-mediated regulation of ABA to activate antiviral immunity against TSWV. Manuscript is well written and results are interesting.

Specific comments:

Authors found that NB domain of Sw-5b NLR interacts with NB domain through Y2H and then confirmed the interaction using split luciferase assay. Authors have also confirmed the interaction between full-length Sw-5b and PP2C only in the presence of NSm effector. Does full length Sw-5b interaction is affected if NB domain is deleted? Also, does auto-active Sw-5b interacts with PP2C in the absence of NSm?

Response: Thanks for this review's insightful thoughts. We have investigated the interaction between Sw-5b without NB domain and PP2C. The results showed that when *NB domain is deleted*, Sw-5b^{Δ1taNB} could not interact with PP2C (see new Supplementary Fig. 4a). We have also examined the interaction between the auto-active Sw-5b D857V mutant and the PP2C protein in the absence of NSm. As shown in new Supplementary Fig. 4b, the auto-active Sw-5b interacted with the PP2C4 protein in the absence of NSm.

Authors show that Nbpp2c knockout increases ABA responsive genes and reduces TSWV infection. Did author test if this effect is specific to TSWV or does it also affect other virus infection?

Response: Thank you for this reviewer's helpful suggestions. To investigate whether this effect was specific to TSWV, the leaves of *Nbpp2c4-5* and *Nbpp2c4-10* mutants were also inoculated with an infectious clone of PVX-GFP and PVY-GFP, respectively, by agroinfiltration. The results showed that both PVX-GFP and PVY-GFP accumulated significantly less in *Nbpp2c4-5* and *Nbpp2c4-10* mutants than those in WT plants (see new Supplementary Fig. 15a–d), suggesting that *Nbpp2c4* knockout also reduces the infection of other viruses.

The data shown clearly demonstrate that the RNAi of SnRK2.3/2.4 enhances TSWV infection and induces more HR spots. Authors show that Sw-5b interferes with the interaction between PP2C and SnRK in the presence of TSWV. Does this interference occur only in the presence of NSm effector?

Response: As shown in new Supplementary Fig. 23a–d, we co-expressed TSWV SR⁽⁺⁾eGFP minireplicon with *NbSnRK2.3/2.4* and *NbPP2C4* in the presence of *EV*, *Sw-5b* or *Sw-5b+NSm* in *N. benthamiana* leaves. The results showed that the co-addition of Sw-5b with only the NSm effector inhibited the accumulation of TSWV SR⁽⁺⁾eGFP minireplicon, these data suggest that Sw-5b NLR releases SnRK2.3/2.4 from PP2C4 inhibition only in the presence of NSm effector.

It is interesting that Sw-5b NLR induces stomatal closure and drought resistance. Does

Sw-5b coexpression with NSm fail to induce stomatal closure in Nbpp2c knockout plants? Is drought resistance in Sw-5b line dependent on NbPP2C?

Response: The drought resistance in Sw-5b line would depend on NbSnRK2.3/2.4 but not NbPP2C. When *NbSnRK2.3/2.4* was knockdown, coexpression of Sw-5b with NSm failed to induce stomatal closure (new Supplementary Fig. 25a–e). The upregulation of ABA response genes by co-expression of Sw-5b and NSm was also compromised by silencing of *NbSnRK2.3/2.4*. However, TRV-mediated systemic silencing of *NbSnRK2.3/2.4* results in plant growth defects and consequently we couldn't test the drought resistance assay.

Reviewer #2 (Remarks to the Author):

The manuscript describes the link between the plant immune NLR receptor Sw-5b and the activation of an ABA-dependent antiviral response against TSWV infection. The authors demonstrate that ABA application has a disease-suppressing effect and that the regulatory PP2C-SnRK2 complex involved in ABA signalling contributes to antiviral immunity. They further show that effector recognition by Sw-5b triggers the binding of Sw-5b to PP2C4 which releases SnrK2 from PP2C4 inhibition and leads to an activation of the ABA-dependent defences. Thus, the authors propose a model in that the Sw-5b mimics the action of ABA receptors that upon ABA binding disrupt the PP2C4-SNRK2 interaction to induce downstream ABA responses.

Together, this manuscript provides interesting mechanistic insight into immune-receptor activation of a hormone signalling pathways and adds to the previously described role of ABA in virus infection. There are, however, several issues that need to be addressed to further substantiate the main conclusions and consider the paper for publication.

#1) Fig. 1d-g/Suppl. Fig.3. Transient expression of the Sw-5b NB domain of full-length Sw-5b together with the TSWV effector NSm triggers ABA signalling and increases ABA levels. To distinguish that Sw-5b induces ABA signalling independent of ABA perception, the expression analysis of ABA response genes as well as TSWV accumulation needs to be done in an ABA deficient background, e.g. by disrupting ABA biosynthesis. The same applies to the experiment done in the tomato system (Fig 1k-n).

Response: Thanks for this reviewer's helpful suggestions. We use tobacco rattle virus (TRV)-mediated gene silencing to knock down *NbABA2* and disrupt the ABA biosynthesis in *N. benthamiana* (Supplementary Fig. 5a–c), followed by co-expression of Sw-5b and NSm. The results showed that despite ABA biosynthesis is disrupted, co-expression of Sw-5b and NSm still significantly up-regulates the ABA response genes (Supplementary Fig. 5d–f). The expression levels of the ABA response genes induced by Sw-5b in the *NbABA2*-silenced plants are lower than those in the non-silenced control plants (see new Supplementary Fig. 5d–f). We also silenced *SLABA2* to inhibit ABA biosynthesis in tomato cv. IVF3545 plants (Supplementary Fig. 8a–c), followed

by TSWV inoculation. TSWV inoculation still up-regulates the ABA response genes in *SLABA2*-silenced plants compared to the mock-inoculated control plants (Supplementary Fig. 8d–f). These data suggest that Sw-5b still induce signaling of the ABA pathway in ABA deficient background, but ABA biosynthesis may have amplified the ABA defense signals.

#2) *Figure 3. It is quite puzzling that silencing of NbSnRK increases the number of HR lesions, which is associated with increased levels of TSWV. How has the virus level been determined, in total leaves or in the lesions? If a proper HR is induced, the virus should not spread beyond the lesions. Hence, it remains completely unclear if the Sw-5b activated ABA-dependent immunity branch is connected to or separated from the cell death-mediated resistance response. Does ABA effect the HR? Can the Sw-5b mediated ABA immunity be uncoupled from the HR? Is it the main component of the Sw-5b mediated resistance, or does it act additively to the cell death response? This part is particularly relevant to clarify experimentally as the previously observed extreme resistance to soybean mosaic virus (SMV) has been coupled to ABA mediated responses but occurs in the absence of HR.*

Response: In contrast to extreme resistance, hypersensitive response (HR) allows virus replicate and movement in certain degree. After virus infects and replicates in the first cell, it moves to the next cell. The virus completes the replication and movement in the new cells. After we see the HR visibly, the virus has already infected hundreds of cells. These hundreds of cells with virus infected undergo HR cell death. When ABA-mediated immunity against TSWV was compromised, some loci of cells become infected by TSWV and start replication and movement cycles, leading to the increase of the number of HR lesions.

Our new results showed that neither ABA treatment nor silencing NbSnRK2.3/2.4 to block ABA pathway affect the cell death (new Supplementary Fig. 20a–d). These data suggest that the Sw-5b mediated ABA immunity is separated from the HR. Several previous studies have also reported that the resistance can be uncoupled from the cell death (Genetics, 2000, 156, 341–350; Proc. Natl. Acad. Sci. U.S.A. 1995, 92, 6597–6601; Mol. Plant Microbe. Interact. 2004, 17, 511–520; Science, 2010, 330, 1393–1397; New Phytol. 2016, 212,161-75)

#3) *Based on the previous publication from the group (Chen et al., 2021, New Phytol.), it remains elusive if the ABA-mediated response is related to the cytoplasmic or nuclear pools of the NLR, in particular as they separate the effects of Sw-5b mediated responses on replication (cytoplasmic) and cell-to-cell/long-distance movement (nuclear). In case of the Rsv-mediated extreme resistance to SMV, ABA seems to effect in particular callose deposition at plasmodesmata to prevent intercellular virus spread. What is the mechanism of ABA-immunity in case of TSWV?*

Response: Our new results showed that both the cytoplasmic and the nuclear pools of the Sw-5b NLR in the presence of NSm can induce ABA response genes (see new

Supplementary Fig. 6a–c). This is consistent with the observation that PP2C4 and PYLs are localized both at the cytoplasm and the nucleus.

We have also investigated the mechanism of ABA-immunity to TSWV. Our results showed that ABA treatment not only block intercellular movement of TSWV but also inhibit the replication of TSWV (new Supplementary Fig. 10a–d).

REVIEWERS' COMMENTS

Reviewer #1 (Remarks to the Author):

The authors have adequately addressed most of the comments raised in the previous review.

Reviewer #2 (Remarks to the Author):

The current manuscript represents a revision of an earlier submission on the link of the plant immune NLR receptor Sw-5b and the activation of an ABA-dependent antiviral response against TSWV infection. Based on the comments of the two reviewers the authors have provided a substantial amount of new data and have addressed most of the issues raised. These additions have considerably strengthened the conclusions of the paper, which now represents an important piece of work with significant impact to the field.